# Machine Learning Algorithms for Prediction of the Quality of Transmission in Optical Networks

**DOI:** 10.3390/e23010007

**Published:** 2020-12-22

**Authors:** Stanisław Kozdrowski, Paweł Cichosz, Piotr Paziewski, Sławomir Sujecki

**Affiliations:** 1Computer Science Institute, Warsaw University of Technology, Nowowiejska 15/19, 00-665 Warsaw, Poland; p.cichosz@elka.pw.edu.pl (P.C.); piotr.paziewski2.stud@pw.edu.pl (P.P.); 2Telecommunications and Teleinformatics Department, Wroclaw University of Science and Technology, Wyb. Wyspianskiego 27, 50-370 Wroclaw, Poland; slawomir.sujecki@pwr.edu.pl

**Keywords:** artificial intelligence, machine learning, optical networks, quality of transmission, machine learning classifiers

## Abstract

Increasing demand in the backbone Dense Wavelength Division (DWDM) Multiplexing network traffic prompts an introduction of new solutions that allow increasing the transmission speed without significant increase of the service cost. In order to achieve this objective simpler and faster, DWDM network reconfiguration procedures are needed. A key problem that is intrinsically related to network reconfiguration is that of the quality of transmission assessment. Thus, in this contribution a Machine Learning (ML) based method for an assessment of the quality of transmission is proposed. The proposed ML methods use a database, which was created only on the basis of information that is available to a DWDM network operator via the DWDM network control plane. Several types of ML classifiers are proposed and their performance is tested and compared for two real DWDM network topologies. The results obtained are promising and motivate further research.

## 1. Introduction

The demand for network bandwidth is constantly growing due to emerging Internet applications such as high-definition video streaming, cloud, 5G, internet of things (IoT), virtual reality, etc. As a consequence, backbone traffic is growing exponentially [1]. In addition, the special circumstances related to Covid-19 prompted many services to move online, thus further increasing the demand for internet access with high bandwidth and quality of service (QoS). Therefore, the backbone network must rely not only on high bandwidth optical links but also on a very good transmission quality. Also, in order to improve the commercial viability, elastic optical networks (EONs) have been proposed to use physical layer resources intelligently and efficiently to increase backbone network capacity and enable dynamic services [2,3,4,5,6,7,8]. Currently, commercial optical network equipment providers offer coherent Dense Wavelength Division Multiplexing (DWDM) technology capable of establishing optical channels with 100 Gigabit per second (Gbps), 200 Gbps, 400 Gbps, and even 1 Tbps throughput [9,10,11,12].

### 1.1. Motivation

Due to continuous traffic growth in backbone networks, dynamic or even programmable optical networks are becoming increasingly important as they allow for more efficient use of network resources without significant increase in operational and capital expenditure (Opex/Capex). In modern DWDM networks, which comply with software defined network (SDN) paradigm, DWDM network reconfiguration is becoming more and more frequent, making the growing network more robust and faster adapting to real changes in the bandwidth demand. Ideally, network reconfigurations should closely match changes in the bandwidth demand. However, bandwidth demand can fluctuate very quickly (fluctuations can occur within minutes), whilst network reconfigurations take much more time and thus reconfigurations that exactly match the bandwidth demand fluctuations are not feasible yet. This is mainly due to operational processes that are too slow to allow for the reoptimization of the network in real time. That is why it is so important that DWDM network reconfiguration is fast, automated and does not incur significant increase in Opex.

Traditional network planning tools estimate Quality of Transmission (QoT) using static Q-factor models, which are functions of the physical layer parameters [13,14]. The extraction of physical layer parameters is not straightforward from the network operator point of view. Further, the models assessing QoT perform complex calculations that are time-consuming, require highly qualified engineers, and purchase of expensive equipment for performing QoT measurements. This obviously significantly increases network operator Opex and in practice is hence avoided.

One of the main goals of telecommunication operators is to minimize DWDM network Capex and Opex by introduction of automation, frequent network reconfiguration, reoptimization and monitoring of network reliability [15]. Currently, a software defined network (SDN) is used to achieve all these goals. SDN uses a logically centralized control plane in a DWDM network, which is built using specially for this purpose flexible hardware, e.g., reconfigurable optical add/drop multiplexers (ROADM), flexible linear interfaces, etc. [16]. However, today’s DWDM networks are growing rapidly in terms of the number of connected devices. This in turn results in increasing complexity of a DWDM network management to a level that is becoming very difficult manage even with SDN platform in place. Hence, there is a need for new ideas and solutions. One of them is the knowledge-based network (KDN) is also a next step on the path towards an implementation of a self-driving network [15,17]. KDN is a complementary solution for SDN that brings reasoning processes and machine learning (ML) techniques into the network control plane to enable autonomous and fast operation and minimization of Opex [18].

One of the key problems in implementing a self-driving optical network is an automatic provisioning of optical channels (lightpaths). The lightpaths accumulate impairments that can degrade channel QoT to such an extent that the transmitted information cannot be accurately extracted at the receiver. Therefore, to quickly deliver a new lightpath or redirect an existing one, for instance in response to a physical link failure, an accurate and quick QoT assessment is required. However, as already mentioned QoT evaluation is associated with complex and time-consuming calculations. Further, the estimation of QoT of an optical channel before its deployment is a step of great importance for optical network operators, which has to be carried out with great caution in order to avoid an immediate link establishment failure [19,20,21,22,23,24,25,26]. Therefore, in this article we propose applying machine learning methods to estimate the QoT of optical channels. It is noted that ML has been already used to address the problem of optical link QoT assessment by other authors [8,15,27,28,29], also for non-coherent networks [30]. However, our approach is based on real data that is easily available to a DWDM network operator through the control plane and does not use data that is only available to DWDM equipment providers and fiber providers. Therefore, the main advantage of the presented approach is that it can be easily implemented by a network operator. As will be explained in the next paragraph, such an approach imposes some constraints on the selection of suitable ML methods, which makes the ML problem considered in this contribution distinctly different from the one addressed in [8,15,27,28,29].

### 1.2. Machine Learning Challenges

Using only real data that is easily accessible through a DWDM network control plane when applying machine learning algorithms to support optical network management poses substantial challenges. These challenges are associated with data representation, data size, and especially with class imbalance.

General-purpose machine learning algorithms are designed to work with tabular data, with each instance or observation represented by a fixed-length vector of a attribute values. Optical networks, or any other graph structures, are not normally represented in such a way and need some feature engineering to be converted to a vector representation. The approach adopted in this work is to consider each network channel as a single instance and to derive a fixed, uniform set of attributes describing all hops in network channels independently of the number of hops.

One of the most important problems in many machine learning applications is the risk of overfitting, i.e., obtaining apparently good models fitting the training data very well, but with poor generalization capabilities. It is an obvious intuition, fully confirmed by the results of computational learning theory [31], that the risk of overfitting increases with the size and complexity of the model space, which is typically directly related to the training data dimensionality (the number of attributes), and decreases with the training data size (the number of instances). Unfortunately, real datasets gathered by network operators tend to be small (hundreds rather than thousands of instances), but the number of attributes needed to adequately describe network channels may be quite large (dozens rather than a few). This is why the experimental study presented in this article uses a set of diverse learning algorithms, with different overfitting prevention mechanisms, and several nested attribute subsets. All these algorithms can be seen to take inspirations from information theory.

Real optical network data for channel classification are more than likely to suffer from severe class imbalance because the number of channels that were not established due to optical impairments (which we will refer to as “bad” in the remainder of the article) is typically less than of those that were allocated (referred to as “good”) by at least an order of magnitude. This is because it is not a standard practice to preserve and archive unsuccessful channel configurations. While this situation may gradually become less severe as network operators become aware of the prospects of machine learning applications to network reconfiguration and the associated data requirements, yet for the foreseeable future one has to deal with datasets where only a tiny fraction of instances represent the minority class. This requires special care when creating predictive models and evaluating their quality. The extreme dominance of one class over the other makes it easy to come up with apparently accurate models with little or no actual predictive utility. To avoid this, in our experimental study we increase the sensitivity of the learning algorithms to the minority class by weighting or altering prior class probabilities, assess prediction quality using ROC and precision-recall curves rather than misclassification error or classification accuracy, and apply the cross-validation procedure with stratified sampling to preserve class distribution and ensure low evaluation bias and variance.

Thus, in this contribution a database is created using only information that is available to a DWDM network operator via the control plane. The data is collected from 2 distinct networks that use equipment coming from 3 different hardware families. Therefore, our intention is to provide an effective tool for assessing QoT in a DWDM network, which can be easily implemented by a network operator. We propose several types of ML classifiers and their performance is tested and compared for the two considered real DWDM network topologies.

### 1.3. Article Organization

The article is organized as follows: in Section 2 we briefly describe the data and analyzed networks. Section 3 provides a detailed description of applied machine learning algorithms, and Section 4 presents the results of the experimental study. Contributions of this work are summarized and future work directions are discussed in Section 5.

## 2. Data and DWDM Network Description

Two DWDM networks have been considered for the application of ML algorithms:The first network consists of 187 nodes, in which a mixture of non-coherent and coherent transponders were installed. Such a network is quite representative of DWDM networks used by operators nowadays whereby legacy non-coherent transponders are still in use whilst modern coherent transponders are gradually introduced. The non-coherent transponders belong to Nokia 1626 family whilst the coherent ones are from Nokia PSS 1830 family. The non-coherent transponders have 2.5G and 10G transmission rate and use NRZ modulation. The coherent transponders operate at either 100G or 200G transmission rate and use three types of modulation: QPSK, 8QAM and 16QAM. In the remainder of the article the dataset corresponding to the 187-node network will be referred to as *dataset 1*.The second network consists of 83 nodes with coherent transponders only. This is a typical representative of a new network established by an operator. In this instance the coherent transponders belong to Ciena’s 6500 family of equipment, with transmission rate of 100G, 200G and 400G and four types of modulation: QPSK, 16QAM, 32QAM and 64QAM. In the remainder of this article the dataset corresponding to the 83-node network will be referred to as *dataset 2*.

The network topology for the 117 node network is presented in Figure 1a while that for the 63 node network in Figure 1b. Both considered networks use 96 DWDM channels allocated in band C and geographically are situated in Poland whereby network nodes correspond to Polish cities. Thus, the network spans approximately an area of 1000 km in diameter whereby the largest distance between two most spatially separated nodes is 120 km and an average distance between two neighboring nodes is about 60 km.

### 2.1. Channel Attributes

Since the aim of the research is to predict QoT for optical channels, the main data object is an optical channel that has a set of attributes. Thus, a table is created where each row corresponds to a channel. The following attributes have been used to characterize each channel:hop_lenghts (a natural number describing the length of the edge, expressed in kilometres, e.g., 67 km);number_of_paths_in_hops (a natural number representing the number of channels in the edge, e.g., 17);hop_losses (a real number describing edge suppression, expressed in decibels e.g., 17.7 dB);number_of_hops (a natural number representing the number of edges of which the path (channel) consists, e.g., 9);transponder_modulation (description of the transponder modulation that is installed at the beginning and the end of the path (channel), e.g., QAM);transponder_bitrate (a natural number describing the transmission rate of the transponder, e.g., 100 Gbps.).

The information about optical channels active in the network is collected from the management system via control plane to a dedicated database. The implemented process of data collection to form a database for machine learning algorithms can be divided into the following stages: (1) generating reports on network parameters from network elements, (2) converting text reports and static tables into appropriate data structures, (3) saving structured data to a database, (4) sharing data with other system modules. Figure 2 illustrates the process of preparing the database for machine learning algorithms.

Figure 3 shows a set of example histograms for optical channel attributes of the two considered data sets. Figure 3a shows how many optical channels we have, consisting of different numbers of hops. The value of the attenuation of individual hops is presented in Figure 3b. Then, Figure 3c shows how many hops of different lengths we have, and the number of hops with different number of optical channels presents the histogram in Figure 3d.

Finally, Table 1 gives the information about the number of the optical channels included in a set that were allocated marked as “good” and the ones that could not be allocated due to low QoT—marked as “bad” (for dataset 1 these numbers are presented separately for non-coherent and coherent transponders). It is clearly seen that all sets considered are unbalanced, i.e., the number of good elements is much larger than the bad ones. This poses a challenge for the ML algorithms, which is discussed in further detail in the next section.

### 2.2. Vector Representation

To transform channel descriptions to a vector representation, an aggregation-based feature engineering technique was applied to channel hops. This technique aggregates each of the available hop properties (hop_lengths, num_of_paths_in_hops, hop_losses) over all hops in a path (i.e., an optical channel) by applying the following set of aggregation functions:mean and standard deviation (assuming 0 for one-hop channels),minimum and maximum,median, the first quartile, and the third quartile,linear correlation coefficient with the ordinal number of the hop in the optical channel.

This yields 8 attributes for each of the 3 hop properties, summing up to 24 attributes derived from hop properties, in addition to the 3 channel attributes unrelated to individual hops (number_of_hops, transponder_modulation, and transponder_bitrate).

## 3. Algorithms

An arbitrary classification algorithm can be used to predict channel “good”/“bad” class labels or probabilities based on available attributes. A selection of the most useful algorithms known from the literature is applied in this work: logistic regression, support vector machines, decision trees, random forests, and extreme gradient boosting [32]. Their main principles of operations as well as properties that make them interesting in the application area investigated by this article are highlighted in the corresponding subsections below.

It is worthwhile to underline that all these algorithms have direct links to information theory. Logistic regression uses the maximum log-likelihood method for parameter estimation. This is also the case for extreme gradient boosting, which is used for binary classification with logarithmic loss. While support vector machines employ a linear-threshold model representation optimized for classification margin maximization rather than an information-theoretic objective function, they are used with Platt’s scaling to obtain probabilistic predictions. The technique transforms the distance from the decision boundary by a logistic function with parameters adjusted for maximum log-likelihood. Finally, the decision tree and random forest algorithms used splits selected to minimize class impurity, typically measured using the entropy or the Gini index.

The support vector machines, random forest, and extreme gradient boosting algorithms belong to the most powerful and often used algorithms for learning from tabular data [33]. This makes them natural and promising candidates for our application domain. The other two algorithms, logistic regression and decision trees, serve as comparison baselines for them, to verify whether simpler and more interpretable model representations are sufficient to achieve a similar level of predictive performance. The increasingly popular deep learning approach [34], that has been spectacularly successful for image, video, audio, or text classification, is not very well suited to tabular data. This is due to the properties of such data (heterogeneous and often sparse and correlated attributes with substantially varying predictive utility, both discrete and continuous, imbalanced classes) and due to the properties of deep learning algorithms (inherent over-parameterization, lack of inductive bias appropriate for tabular data, high computational demands). Examining the utility of recent deep learning architectures specifically designed for tabular data that overcome these limitations [35,36] is postponed to future work.

### 3.1. Logistic Regression

Logistic regression is an instantiation of generalized linear models which adopts a composite model representation function, with an inner linear model and an outer *logit* transformation [37]. Training a logistic regression model consists in finding model parameters which maximize the log-likelihood of training set classes.

Due to the probabilistic objective function used for parameter estimation, logistic regression can generate well-calibrated probability predictions and is often the classification algorithm of choice where this is required. It is easy to apply and not overly sensitive to overfitting unless used for high-dimensional data. In our experiments, logistic regression serves as a natural comparison baseline for the more refined support vector machines algorithm which extends linear classification, achieving better overfitting resistance and permitting nonlinear relationships.

### 3.2. Support Vector Machines

Support Vector Machines (SVM), which often belong to the most effective general-purpose classification algorithms, can be viewed as a considerably strengthened version of a basic linear-threshold classifier with the following enhancements [38,39,40]:**margin** **maximization:**the location of the decision boundary (separating hyperplane) is optimized with respect to the classification margin,**soft** **margin:**incorrectly separated instances are permitted,**kernel** **trick:**complex nonlinear relationships can be represented by representation transformation using kernel functions.

The SVM algorithm assumes a binary classification scenario with two classes. Class predictions are generated using a standard linear-threshold rule. Model parameters are found by solving a quadratic programming problem defined to achieve classification margin maximization, i.e., placing the decision boundary so as to maximize the distance from the closest correctly separated instances, with a penalty for constraint violations controlled by a cost parameter. Non-linear relationships can be represented by an implicit input transformation using kernel functions. The algorithm is sensitive to the settings of the cost parameter as well as the kernel function type and its parameters – these may need to be tuned for the best predictive performance.

Binary linear-threshold SVM predictions are determined based on whether an instance lies on the positive or on the negative side of the separating hyperplane. Probabilistic predictions are obtained by Platt’s scaling—applying a logistic transformation to the signed distance of classified instances from the decision boundary, with parameters adjusted for maximum likelihood [41].

A noteworthy property of SVM is the insensitivity of model quality to data dimensionality, which—unlike for many other algorithms—does not increase the risk of overfitting because model complexity is related to the number of instances close to the decision boundary rather than to the number of attributes.

### 3.3. Decision Trees

A decision tree [42,43] is a hierarchical structure that represents a classification model. Internal tree nodes represent splits applied to decompose the domain into regions, and terminal nodes assign class labels or class probabilities to regions believed to be sufficiently small or sufficiently uniform.

Decision trees are popular in many applications due to their capability of combining reasonably good prediction accuracy with the human readability of models. They may require appropriately tuned stop criteria or pruning to avoid overfitting. In our experiments, decision trees serve as a natural comparison baseline for the more refined random forest and extreme gradient boosting algorithms which combine multiple trees to achieve better prediction quality and overfitting resistance.

### 3.4. Random Forest

Random forests belong to the most popular ensemble modeling algorithms [44], which achieve improved predictive performance by combining multiple diverse models for the same domain. A random forest [45] is an ensemble model represented by a set of unpruned decision trees, grown based on multiple bootstrap samples drawn with replacement from the training set, with randomized split selection. It can be considered an enhanced form of bagging [46], which additionally stimulates the diversity of individual models in the ensemble by randomizing the decision tree growing algorithm used to create them.

Random forest prediction is achieved by simple unweighted voting of individual trees from the model. Vote distribution can be also used to obtain class probability predictions. With sufficiently many diversified trees (typically hundreds) this simple voting mechanism usually makes random forests extremely accurate and resistant to overfitting. As a matter of fact, in many cases they belong to the most accurate classification models that can be achieved. The random forest algorithm is not overly sensitivity to parameter settings, which makes it easy to use and capable of producing high quality models without excessive tuning.

An additional capability of the random forest algorithm is providing measures of attribute predictive utility, referred to as variable importance. The most reliable of those is based on the decrease of prediction accuracy resulting from random attribute value permutation, estimated using out-of-bag training instances [47]. This measure is used in the experiments reported in this article.

### 3.5. Extreme Gradient Boosting

The extreme gradient boosting or *xgboost* algorithm is another ensemble modeling algorithm that has gained high popularity and turned out highly successful in many applications. It belongs to the family of boosting algorithms the main principle of which is to create ensemble components sequentially in such a way that each subsequent model best compensates the imperfections of the previously created ones [48,49]. Being an enhanced version of gradient boosting machines [50,51], *xgboost* is a unified algorithm for classification and regression that internally uses regression trees for model representation and creates trees so as to optimize an ensemble quality measure that includes a loss term representing the level of fit to the training data and a regularization term penalizing model complexity [52].

Extreme gradient boosting applied to binary classification is typically used with logarithmic loss, which is the negated log-likelihood of training set classes. The numeric predictions of individual trees are summed up transformed by a logistic link function to obtain class probability predictions.

The extreme gradient boosting algorithm usually delivers excellent prediction quality, sometimes superior to that obtained by random forest models. It can overfit, however, if the number of trees grown is too large and therefore requires tuning at least of this parameter.

### 3.6. Increasing Sensitivity to the Minority Class

The datasets used in our experimental studies suffer from an extreme form of class imbalance. This may cause algorithms designed to minimize the misclassification error produce apparently highly accurate but useless models that fail to correctly classify most instances of the minority class. Most of the algorithms described above have mechanisms for increasing the sensitivity of the minority class that help one to avoid this pitfall.

The SVM algorithm can be made more sensitive to the minority class by specifying class weights, applied when calculating the penalty for constraint violations. Higher penalties for the minority class make the algorithm find a separating hyperplane that classifies these instances correctly and with a high margin, as far as possible.

A standard way of dealing with class imbalance in decision tree modeling is to use instance weights to make the algorithm more sensitive to the minority class. An equivalent effect can be also achieved by specifying a prior class distribution, to be used when selecting splits, checking stop criteria, and determining leaf classes and probabilities instead of the actual unbalanced class distribution observed in the training data. The latter approach is used in the experiments reported in this article.

The random forest algorithm, as most ensemble modeling algorithms, tends to be more robust with respect to class imbalance. There are two mechanisms that may still sometimes improve model quality when learning from highly unbalanced training set. One is stratified sampling applied to drawing bootstrap samples, with different selection probabilities for particular classes. In the extreme case, a bootstrap sample may contain all instances from the minority class and the sample of the same size from the dominating class. Another approach is to specify instance weights, similarly as for decision trees. Since in our case classes are extremely imbalanced and there are only a few instances of the minority class, the weighting technique is preferred to the stratified sampling technique, since the latter would have to severely undersample the dominating class, with a possibly negative effect on model performance.

The logistic regression and *xgboost* algorithms which minimize the logarithmic loss rather than the misclassification error, are generally resistant to the impact of class imbalance as long as class probability predictions are used rather than class label predictions. This is the case in our experiments, where prediction quality is evaluated using ROC and precision-recall curves. When class label predictions are needed, an appropriate probability cutoff threshold can be determined (e.g., corresponding to some selected operating point on the ROC or PR curve) and used instead of the default 0.5 threshold, which would only make sense for balanced classes. With that being said, the extreme gradient boosting algorithm can use user-specified instance weights when growing trees, and this mechanism for increasing its sensitivity to the minority class is used in our experimental study.

### 3.7. Classification Model Evaluation

The most common classification quality measures such as the misclassification error or classification accuracy are not very useful whenever classes are unbalanced or likely to have different predictability. This is why a quality measure sensitive to misclassification distribution is required. In the experiments reported in this article classification quality is visualized using ROC curves, presenting possible tradeoff points between the *true positive rate* and the *false positive rate* [53,54]. The former is the share of instances of the positive class which are correctly predicted to be positive and the latter is the share of instances of the negative class which are incorrectly predicted to be positive. The performance across all possible tradeoffs can be summarized using the area under the ROC curve (AUC). The decision which of the two classes is considered positive and which is considered negative is arbitrary to some extent, but it is a common convention to consider positive the class that is less frequent and harder to predict—in our case, it is the “bad” class.

The area under the ROC curve can be interpreted as the probability that a randomly chosen positive instance has a higher predicted probability of the positive class than a randomly chosen instance of a negative class. Random guess and constant predictions both correspond to an AUC value of 0.5, indicating no predictive power. This clear interpretation is an additional benefit of the ROC analysis. Under severe class imbalance, however, with the vast majority of instances being negative, the false positive rate will not decrease substantially even if a large share of positive class predictions is incorrect, because the number of false positives may be still small relative to the negative class count. This is why it may be reasonable to additionally consider the *precision*, which is the share of positive class predictions that are correct, and examine its tradeoff with the *recall*, which is another term for the true positive rate. The range of possible tradeoffs is then visualized by precision-recall (PR) curves and can be summarized by the area under the PR curve (PR AUC), which can be interpreted as the average level of precision achieved across the whole range of recall values.

To achieve reliable, low-bias and low-variance predictive performance estimates, the n×k-fold cross-validation procedure times is applied [55]. It makes an effective use of the available data for both model creation and evaluation by randomly splitting it into *k* equally sized subsets, each of which serves as a test set for evaluating the model created on the combined remaining subsets, and repeating this process *n* times to further reduce the variance. The true class labels and predictions for all n×k iterations are then combined to determine ROC curves, PR curves, and the corresponding AUC values. Due to the severe class imbalance, the random partitioning into *k* subsets is performed by stratified sampling, preserving roughly the same number of minority class instances in each subset. We use k=5 for dataset 1 (with 15 “bad” channels) and k=3 for dataset 2 (with 3 “bad” channels), and n=50 for both the datasets.

## 4. Experiments

This section presents computational experiments in which the algorithms described in Section 3 are applied to the two datasets described in Section 2. The primary objective of the experiments is to assess the best level of prediction quality possible to achieve for channel classification using real optical network data. It is also interesting to observe how well particular algorithms cope with the challenges of small and extremely unbalanced datasets. Attribute predictive utility is also measured to gain possibly useful insights about which channel properties have the highest impact on the predicted class.

### 4.1. Attribute Subsets

Since the small data size with many attributes increases the risk of overfitting and not all attributes may be required for successful channel classification, the following nested attribute subsets are used:**subset** **0:**number_of_hops, transponder_modulation, and transponder_bitrate,**subset** **1:**subset 0 plus all attributes obtained by applying the mean and standard deviation aggregation functions to hop properties,**subset** **2:**subset 1 plus all attributes obtained by applying the minimum and maximum aggregation functions to hop properties,**subset** **3:**subset 2 plus all attributes obtained by applying the median, first quartile, and third quartile aggregation functions to hop properties,**subset** **4:**subset 3 plus all attributes obtained by applying the correlation aggregation function to hop properties (i.e., the full attribute set).

Determining which attribute subsets works best for particular algorithms is a part of our configuration tuning process.

### 4.2. Algorithm Implementations and Setup

The following algorithm implementations are used in the experiments:**logistic** **regression:**the implementation provided by the standard glm R function [56],**SVM:** the implementation provided by the e1071 R package [57],**decision** **trees:**the implementation provided by the rpart R package [58],**random** **forest:**the implementation provided by the ranger R package [59],**extreme gradient** **boosting:**the implementation provided by the xgboost R package [60].

Since the *xgboost* algorithms does not directly support discrete attributes and one attribute in the dataset is discrete, it was preprocessed by converting particular discrete values to binary indicator columns.

For the logistic regression and SVM algorithms parameters controlling the underlying optimization process were left at default values and the radial kernel was used. The following SVM parameters specifying the optimization problem were tuned by grid search:cost:the cost of constraint violation,gamma:the kernel parameter,class.weights:class weight for the minority class in constraint violation penalty (a weight of 1 was used for the dominating class).

For the decision tree algorithm uniform prior probabilities for the two classes were set via the prior parameter. Parameters specifying the stop criteria were tuned by grid search:minsplit:the minimum number of instances required for a split,cp:the complexity parameter,maxdepth:the maximum tree depth.

For the random forest algorithm, the following parameters were tuned by grid search:num.trees:the number of trees,mtry:the number of attributes for split selection at each node,case.weights:weights for instances of the minority class (weights of 1 were used for instances of the dominating class).

For the extreme gradient boosting algorithm parameters controlling the boosting process were left at default values except for the following settings, tuned by grid search:nrounds:the number of boosting iterations,weight:weights for instances of the minority class class (weights of 1 were used for instances of the dominating class).

It is worthwhile to notice that the parameter setups for the SVM, decision tree, random forest, and extreme gradient boosting algorithms include settings responsible for properly handling unbalanced classes (ensuring sufficient sensitivity to the minority class), as discussed in Section 3.6. This is achieved by specifying class weights for SVM (assigning a higher weight to the minority class when calculating the constraint violation penalty term in the optimization objective), setting uniform class priors for decision trees, and specifying higher minority class instance weights for the random forest and *xgboost* algorithms. These settings were verified to indeed improve model quality. No form of class rebalancing is necessary for the logistic regression algorithm, since any class weights or priors would only shift the default class probability cutoff point used for predicted class label assignment. This would serve no useful purpose given the fact that the ROC analysis used for predictive performance evaluation is based on predicted class probabilities instead of class labels anyway.

Since from preliminary experiments we found that precision-recall curves were much more sensitive to attribute subsets and algorithm parameter setups than ROC curves, we adopted the following tuning procedure with preference for high PR AUC:apply each algorithm with all attribute subsets and all grid search parameter setups,select the attribute subset with the highest maximum PR AUC over all parameter setups,select the parameter setup with both a near-maximum PR AUC and a near-maximum ROC AUC for the previously selected attribute subset, where ‘near-maximum’ was technically interpreted as ‘at least 0.99 of the maximum’.

### 4.3. Results

Table 2 displays the best configurations identified by the tuning process. It is interesting to notice that for dataset 1 the random forest algorithm was able to successfully use the biggest attribute subset 4, the SVM and *xgboost* worked best with the medium subset 2, whereas logistic regression and decision trees achieved their best performance with the smallest subset 0. This is quite consistent with the level of overfitting resistance of particular algorithms, with the random forest known to have the lowest and decision trees known to have the highest risk of overfitting. For dataset 2, however, all algorithms worked best with small attribute subsets: subset 1 for random forest and subset 0 for all the others. This suggests that the two datasets have not only different sizes and class distributions, with dataset 2 data being smaller and more unbalanced, but also different complexity of relationships between classes and attribute values. These relationships are likely to be more complex for dataset 1 than for dataset 2, if more attributes are needed to capture them for the former than for the latter. The difference between the datasets is also reflected by algorithm parameter settings. The more heavily unbalanced dataset 2 requires larger minority class weights for the SVM, random forest, and *xgboost* algorithms.

Figure 4 presents the ROC curves obtained for the best identified attribute subset and algorithm configurations. One can observe that:**for dataset** **1:**the prediction quality achieved by all the algorithms appears to be very good, with AUC values between 0.86 and 0.89,reasonable model operating points are possible, with the true positive rate of 0.9 or more and the false positive rate of 0.2 or less,the random forest, *xgboost*, and logistic regression algorithms achieve the best predictive performance, followed by SVM and decision trees,**for dataset** **2:**all algorithms appear to achieve nearly perfect predictions, with AUC values of 0.97–0.98,nearly perfect model operating points are possible, with the true positive rate of 1 and the false positive rate of 0.05 or less,the logistic regression, decision trees, random forest, and *xgboost* peform on roughly the same level, and SVM is only marginally worse.

As discussion in Section 3.7, ROC curves may not provide a sufficient picture of model performance under severe class imbalance, because even with many false positives the false positive rate remains small due to the dominating overall negative class count. Precision recall curves, presented in Figure 5, do indeed show a more useful view of the predictive power of models produced by particular algorithms. The following observations can be made:**for dataset** **1:**the logistic regression, decision trees, and SVM algorithms fail to achieve an acceptable level of precision, with the PR AUC below 0.1,the random forest and *xgboost* algorithms produce much more useful models, with the average precision above 0.4,even for the best random forest models there is the precision drops quickly when recall exceeds 0.4,**for dataset** **2:**all algorithms manage to achieve average precision of about 0.4 or more,a reasonable level of precision can be maintained over a wide range of recall values,the *xgboost* and random forest algorithms achieve the best predictive power, and decision tree models are the worst.

From a practitioner’s point of view, it may be interesting to see how essential parameter tuning was for particular algorithms. Table 3 compares the AUC and PR AUC values of the decision tree, SVM, random forest, and extreme gradient algorithms with default and tuned settings (for *xgboost* there is no default setting for the nround parameter, so we arbitrarily used 50 as the default). It is easy to see that parameter tuning improved the results substantially for all algorithms. Even for the random forest algorithm, which is not overly sensitive to parameter settings, it was beneficial (particularly on dataset 2), and it turned out absolutely necessary for SVM to deliver acceptable results.

Figure 6 presents the random forest variable importance values, with attributes ranked from the most useful at the top to the least useful at the bottom. It can be seen that:**for dataset** **1:**the most useful attributes are number_of_hops, the minimum of num_of_paths_hops, as well as attributes derived by aggregating hop_lengths and hop_losses, whereas transponder_modulation and transponder_bitrate have little or no predictive utility,**for dataset** **2:**transponder_modulation and transponder_bitrate are the most useful attributes by far.

### 4.4. Discussion

The presented results confirm that optical network channel classification using real data is a challenging learning task. When looking at ROC curves the level of predictive performance might appear very good or near perfect, with little differences between algorithms, all of which appear fully capable of delivering predictions with high true positive rates and low fall positive rates. However, precision-recall curves reveal that false positive predictions are actually quite frequent, leading to precision values of around 0.4 at best. They also show that the choice of algorithms and parameter settings does indeed matter a lot. Since the datasets are small and highly unbananced, special care is needed to prevent overfitting and ensure sufficient sensitivity to the minority class.

The performance level achieved for the best identified configurations for the two datasets is definitely useful. The AUC values approaching or exceeding 0.9 are high above the random guess level of 0.5, and the average precision (the area under the precision-recall curve) of about 0.4, while not perfect, is actually more than satisfactory given the extremely low share of “bad” channels in the training data. Actually, it is not only the small share, but also the small absolute number of positive instances that prevents learning algorithms from creating more successful models. While the skewed class distribution can be compensated for by weighting or setting prior class probabilities, just a handful of training instances provides very poor basis for detecting generalizable patterns.

## 5. Conclusions

A set of machine learning algorithms have been applied to DWDM optical networks in order to estimate the QoT for an optical channel. The ML algorithms have been applied to the data sets derived solely from the DWDM network management layer via the control plane. Thus the proposed approach can be fairly easily implemented by a network operator. The obtained ROC curves suggest near perfect predictive performance of all implemented ML algorithms. However, precision-recall curves reveal that the network channel configuration task with small and unbalanced data is indeed quite challenging, and the choice of algorithms and their configurations matters.

Of the applied algorithms, random forest and extreme gradient boosting delivered the best level of predictive performance. While logistic regression, decision trees, and SVM achieved similar levels of tradeoff between the true positive rate and the false positive rate, they were less successful in maintaining an acceptable level of precision. Each algorithm benefited from parameter tuning, although random forest—as expected—worked reasonably well with default settings.

Model operating points were obtained with more than 90% of “bad” channel configurations correctly detected, less than 20% of “good” configurations incorrectly predicted to be “bad”, and at least 40% of channel configurations predicted to be “bad” are actually bad. This level of classification quality—promising but leaving space for improvement—provides a strong encouragement for future research on QoT prediction using machine learning.

One obvious but possibly the most useful future work direction is to gather more data, particularly including more instances of “bad” channels. This would provide more space for detecting generalizable, predictively useful relationship patterns, and make model evaluation results more reliable and convincing. With that being stated, there are still some interesting enhancements of the modeling procedure possible with the currently available data. One of those is augmenting the datasets with artificially generated minority class instances, e.g., using the SMOTE technique [61].

It may be also interesting to combine binary classification models, such as those presented in this article, with one-class classification models that can be learned from unlabeled data containing exclusively or mostly instances of a single class. One-class classification algorithms that could be considered include one-class SVM [62], isolation forest [63], and one-class random forest [64].

## Figures and Tables

**Figure 1 entropy-23-00007-f001:**
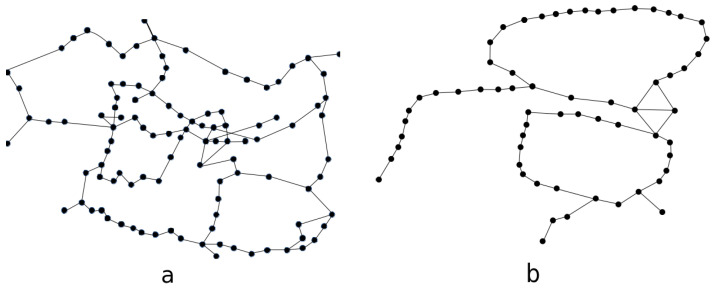
Analyzed network topology for (**a**) 117 node network, and (**b**) 63 node network topologies.

**Figure 2 entropy-23-00007-f002:**
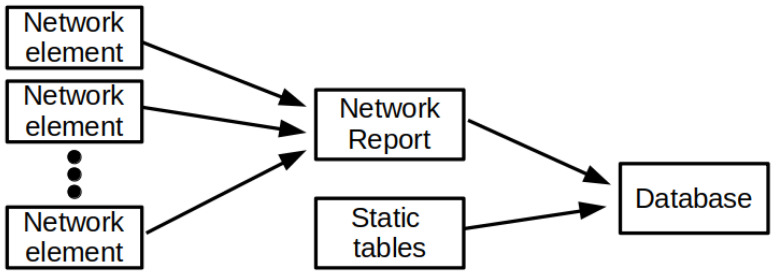
Data preparation process.

**Figure 3 entropy-23-00007-f003:**
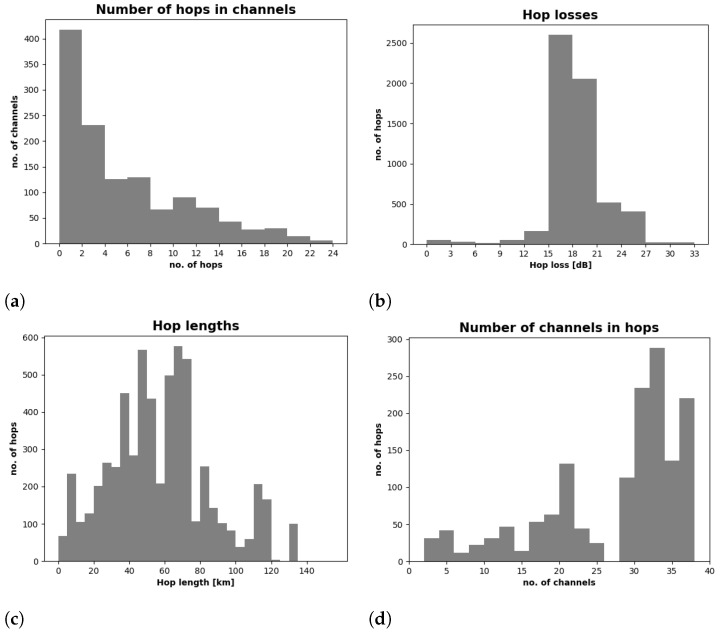
Histograms of optical channel attributes for all data sets under consideration.

**Figure 4 entropy-23-00007-f004:**
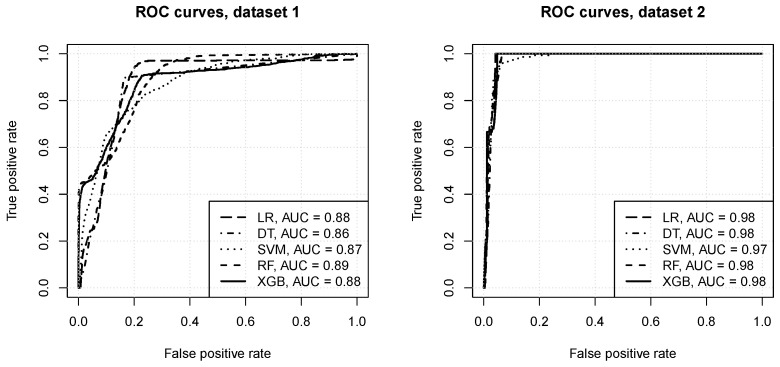
The ROC curves.

**Figure 5 entropy-23-00007-f005:**
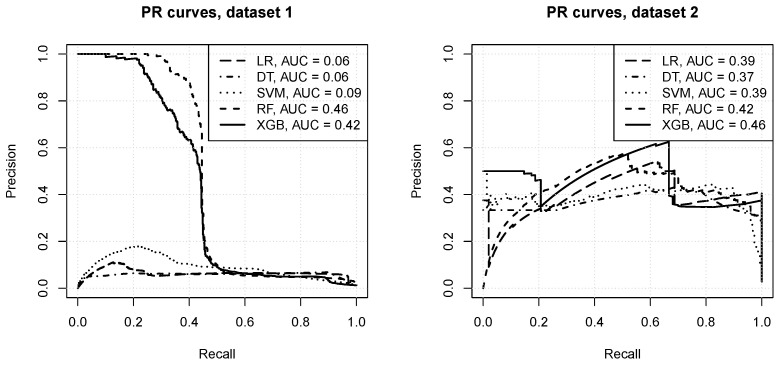
The precision-recall curves.

**Figure 6 entropy-23-00007-f006:**
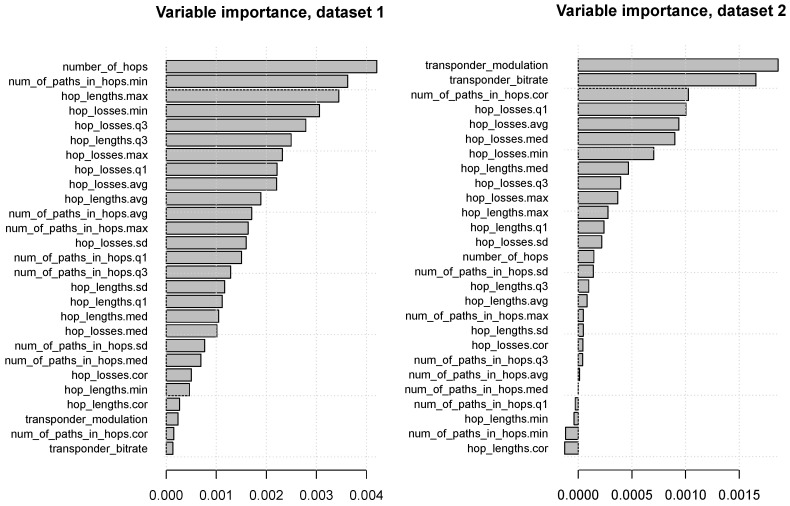
The random forest variable importance plots.

**Table 1 entropy-23-00007-t001:** Data characteristics for the considered data sets.

**Dataset 1**	
**Optical Channel Type**	**#Channel**
“good” non-coherent optical channels	1046
“good” coherent optical channels	87
“bad” non-coherent optical channels	13
“bad” coherent optical channels	2
**Dataset 2**	
**Optical Channel Type**	**#Channel**
“good” coherent optical channels	103
“bad” coherent optical channels	3

**Table 2 entropy-23-00007-t002:** Attribute subsets and algorithm configurations identified by tuning.

**Dataset 1**		
**Algorithm**	**Attribute Subset**	**Parameter Settings**
Logistic regression	subset 0	—
Decision trees	subset 0	minsplit=10, cp=0.001, maxdepth=2
SVM	subset 2	cost=10, gamma=0.15, class.weights=5
Random forest	subset 4	num.trees=500, mtry=10, case.weights=2
*xgboost*	subset 2	nrounds=5, weight=5
**Dataset 2**		
**Algorithm**	**Attribute subset**	**Parameter settings**
Logistic regression	subset 0	—
Decision trees	subset 0	minsplit=15, cp=0.0002, maxdepth=4
SVM	subset 0	cost=5, gamma=0.15, class.weights=20
Random forest	subset 1	num.trees=1000, mtry=2, case.weights=20
*xgboost*	subset 0	nrounds=15, weight=20

**Table 3 entropy-23-00007-t003:** Results for default and tuned parameter settings.

**Dataset 1**				
**Algorithm**	**Default Parameters**		**Tuned Parameters**	
	**AUC**	**PR AUC**	**AUC**	**PR AUC**
Decision trees	0.77	0.05	0.86	0.06
SVM	0.74	0.06	0.87	0.09
Random forest	0.90	0.43	0.89	0.46
*xgboost*	0.88	0.20	0.88	0.42
**Dataset 2**				
**Algorithm**	**Default Parameters**		**Tuned Parameters**	
	**AUC**	**PR AUC**	**AUC**	**PR AUC**
Decision trees	0.94	0.31	0.98	0.37
SVM	0.58	0.09	0.97	0.39
Random forest	0.97	0.29	0.98	0.42
*xgboost*	0.97	0.30	0.98	0.46

## Data Availability

Data sharing not available.

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
