# Peer review of "Machine Learning Algorithms for Prediction of the Quality of Transmission in Optical Networks"

_entropy, 2020, doi:10.3390/e23010007_

Round 1
Reviewer 1 Report
The paper contains innovative contributions but some changes/clarifications are needed;
1)The Channel Attributes should be defined more clearly (Pag. 4);
2)The choice of the machine learning algorithms should be motivated; the authors use traditional algorithms; conversely more sophisticated algorithms, based on deep learning, have been recently defined; at least one of these innovative algorithms could be used for the prediction;
3)More information about the considered networks.
4)Other real networks [1] could be considered to prove the effectiveness of the proposed solution; the authors should be able to generate the database for these networks.
[1] F. Matera, V. Eramo, A. Schiffini, M. Guglielmucci, and M. Settembre, “Numerical Investigation on Design of Wide Geographical Optical-Transport Networks Based on n 40-Gb/s Transmission,” IEEE/OSA Journal Lightwave of Technology, vol. 21, pp. 456-465, 2003.
Author Response
Dear Reviewer,
We are very grateful to the Reviewer for taking time to review the manuscript and providing comments that give us a chance to improve its quality.
Please find attached file with our detailed answers.
Kind Regards,
Stanislaw Kozdrowski

Reviewer 2 Report
The content of the article fits perfectly into the scope of a Entropy journal. There is no doubt that the article deserves to be published, especially in a section Information Theory, Probability and Statistics.
One of the reasons for this is that a key research topic includes issues of improving the quality of transmission (QoT) in optical networks with the coherent Dense Wavelength Division Multiplexing (DWDM) technology offered by operators. This is very important in view of the constantly growing demand for network bandwidth, especially in the current situation with Covid-19.
The authors have proposed a method of QoT assessment for the optical channel in DWDM optical networks based on a set of machine learning algorithms. The advantage is that this method can be fairly easily implemented by a network operator. It is relevant and interesting.
The 30 publications analyzed in the introduction of an article were the basis for achieving the above-mentioned goal.
The results of relevant computational experiments are given and explained in order to estimate the best possible QoT level for channel classification using real optical network data. The article contains some new data.
In the end of the article the most useful directions for future research are presented, consisting in collecting more data and enhancements the modeling procedure, especially using Synthetic Minority Over-sampling Technique, combining binary classification models with single-class classification models including one-class Support Vector Machines, isolation forest, and one-class random forest.
The text is clear and easy to read.
The “Conclusions” are consistent with the evidence and arguments presented and address to the main question posed.
NOTE: The article is presented in logical way and overall written well but:
- in the Section “Conclusions” are provided references to literature sources, which is not specific to this section. These data would have to be transferred to the main part of the article, while the Section “Conclusions” would have included general and comparative results;
- it needs editorial improvement (e.g. in a line 381 “decision trees: the implementation provided by the rpart R package [55],“.
Author Response

(The authors gave the same response as above.)

Reviewer 3 Report
Thank you for this interesting and strong work.
I have few comments (Minor) to be considered.
1) In "Abstract", you used DWDM and ML without definition (found in "Introduction")... In the first appearance, in general, it is preferred to write every word and then the abbreviation.
2) At the end of the first paragraph in "Introduction", you write 100G, 200G, 400G and even 1T ..... is it G bit per second or What? Please, clarify. Also, please, leave a space between any value and its units.
3) The methodology is too long.... If you see you can shorten it (deleting some details which are not important), so make this. Otherwise, leave it as it is.
4) The same note also is in the "Conclusion"... It is too long. We focus only on the main results (including some numerical values as you mentioned).
5) In Fig. 3, it is better to write no. of ????? in both x-axis and y-axis captions than the symbol #.
6) Some figures need to be colored (like Fig. 4 and Fig. 5).
7) In the "References" list, try to be consistent in writing references details (including journal name).... Ex: Ref. 6 and Ref. 7 have the same journal name.... but written in different ways!
8) When the reference is a conference, you must mention the place where the conference was held (city+country). Ex: Ref. 7 and others.
Best Wishes
Author Response

(The authors gave the same response as above.)

Round 2
Reviewer 1 Report
All of the suggested changes have been addressed.
This manuscript is a resubmission of an earlier submission. The following is a list of the peer review reports and author responses from that submission.